# Plant-soil feedbacks help explain biodiversity-productivity relationships

Leslie E. Forero [1], Andrew Kulmatiski [1✉], Josephine Grenzer [1] & Jeanette M. Norton [2]

Species-rich plant communities can produce twice as much aboveground biomass as monocultures, but the mechanisms remain unresolved. We tested whether plant-soil feedbacks (PSFs) can help explain these biodiversity-productivity relationships. Using a 16-species, factorial field experiment we found that plants created soils that changed subsequent plant growth by 27% and that this effect increased over time. When incorporated into simulation models, these PSFs improved predictions of plant community growth and explained 14% of overyielding. Here we show quantitative, field-based evidence that diversity maintains productivity by suppressing plant disease. Though this effect alone was modest, it helps constrain the role of factors, such as niche partitioning, that have been difficult to quantify. This improved understanding of biodiversity-productivity relationships has implications for agriculture, biofuel production and conservation.

---

[1] Department of Wildland Resources and the Ecology Center, Utah State University, Logan, UT, USA. [2] Department of Plants, Soils and Climate, Utah State University, Logan, UT, USA. ✉email: andrew.kulmatiski@usu.edu

Plant productivity typically increases with species richness[1–3]. Efforts to understand this fundamental aspect of ecosystem function (i.e. overyielding) have understandably focused on mechanisms of overyielding, such as complementarity and selection effects[4,5]. Complementarity effects are often attributed to niche partitioning, which allows species-rich communities to capture more resources than species-poor communities[6]. Selection effects occur when more productive species are over-represented in species-rich relative to species-poor communities. However, niche complementarity and selection effects do not fully explain biodiversity-productivity relationships[7–9]. For example, while most plant communities overyield, some communities underyield and niche-partitioning and selection effects generally do not help explain this wide range of responses[2].

Plant-soil interactions offer the potential to explain both overyielding and underyielding[10–12]. For example, species-specific soil pathogens can be expected to be more abundant and decrease plant growth more in monocultures than species-rich communities resulting in overyielding[3,13–15]. Conversely, species-specific soil mutualists can be expected to be more abundant and increase plant growth more in monocultures than species-rich communities resulting in underyielding[16,17]. Although it is near-impossible and likely inappropriate to individually characterize the effect of each species-specific soil pathogen and mutualist on plant productivity, it is possible to summarize the net effect of negative and positive plant-soil interactions using plant-soil feedback (i.e. PSF) experiments[18]. Thus, PSFs offer the potential to help explain both overyielding and underyielding in biodiversity-productivity relationships[10,14].

Several experimental approaches have been used to explore the role of plant-soil interactions in biodiversity-productivity relationships[5,14,19–23]. Perhaps the best support comes from field and potted studies that used fungicide[13] and microbial inoculations[24] to demonstrate soil organism effects on biodiversity-productivity relationships, but these types of sterilization and inoculation experiments have been found to exaggerate PSF effects[24,25]. Several studies have used greenhouse experiments[10,14,22,26], but greenhouse experiments have been found to produce PSFs that are not correlated with field-measured PSF[27].

Two-phase, factorial field experiments remain the preferred approach for describing PSF[28–31]. In these experiments, plant species are grown on soils trained by other plant species. Due to the sample sizes required by factorial designs, these experiments have rarely been performed with more than a few species in the field[19,32]. We are not aware of any two-phase, field experiments that have tested the effects of PSF in biodiversity-productivity relationships.

Our goal was to test whether field-measured PSFs could help predict plant growth in experimental plant communities with 1–16 plant species. To do this, we measured PSFs in the field using a two-phase experiment in which each species was grown on soils trained by each other species in the experiment. This produced growth rates for each species on each soil training type that were used in plant-community simulation models to predict how much biomass each species would produce in a plant community. Plant communities with a range of species richnesses (1–16 species) were grown separately. Model prediction were compared to observed species biomass. To better understand how PSFs affected model predictions, the model was executed either with (PSF simulation model) or without (Null simulation model) PSF effects. To better understand the mechanisms determining how community biomass changes across species richness levels, we separated net biodiversity effects in each dataset (observed, Null, PSF) into complementarity and selection effect

components[11,33]. Because this experiment produced 240 PSF values, it was also possible to test if PSF changed with phylogenetic distance which, if found, would help generalize PSF effects in the biodiversity-productivity relationship[34,35].

## Results

**PSF experiment**. The PSF experiment was performed, primarily, to produce plant growth rates on different soil training types to be used in plant-community simulation models, but we also report PSF index values because they are a common metric that provide a simple summary of plant-soil interactions[29]. PSF index values are the biomass on 'self' soils minus biomass on 'other' soils divided by the maximum of biomass on 'self' or 'other' soil. After a two-year training phase, plants created soils that changed subsequent plant growth by 27% (i.e. the mean absolute value of the PSF index was 0.27 in 2018). However, because most PSFs were negative but some were positive, the net effect of all PSFs was that plants created soils that decreased plant growth by 10% (i.e. a PSF index value of −0.10 in 2018). These effects increased over time during Phase II. The absolute value of the PSF index increased from 0.23 in 2017 to 0.27 in 2018 ($T_{239} = −3.1$, $P = 0.002$). The net value of PSF index values increased from 0.00 in 2017 to −0.10 in 2018 ($T_{239} = 5.4$, $P < 0.001$).

At the species*soil-level, 23 PSF index values were negative, and 13 were positive (i.e. 95% confidence interval did not overlap zero; Fig. 1a). These 36 PSF index values occurred across species so that 14 of 16 species demonstrated a PSF on at least one soil cultivation type (Fig. 1b). For conciseness, only 2018 species*soil-level PSF index values are shown in Fig. 1a. PSF index values among C3 grasses ($T_{1,59} = 5.30$, $P < 0.001$) and forbs ($T_{1,59} = 3.65$, $P < 0.001$) were negative, but PSF index values among C4 grasses ($T_{1,59} = 0.65$, $P = 0.26$) and legumes ($T_{1,59} = 0.73$, $P = 0.23$) were neutral. However, there was no difference among functional groups ($F_{3,224} = 1.58$, $P = 0.19$). There was also no correlation between species*soil-level PSF index values and phylogenetic distance ($F_{1,239} = 0.01$, $P = 0.91$).

When species*soil-level PSF index values were averaged across soil cultivation types to produce one PSF index value for each species, there were five negative and three positive species-level PSF index values in 2017 and five negative and one positive species-level PSF index values in 2018 (Fig. 1b).

**Biodiversity-productivity experiments**. After 4 years, community biomass in the 2014 biodiversity-productivity experiment increased with species richness (Fig. 2; $F_{1,59} = 36.4$, $P < 0.001$) from 55.6 g m$^{-2}$ in monocultures to 187.3 g m$^{-2}$ in 16-species communities (Fig. 2). This 131.8 g m$^{-2}$ difference represented a 237% increase in biomass production. Complementarity effects explained 172.5 g m$^{-2}$ overyielding and selection effects explained 40.8 g m$^{-2}$ underyielding in 16-species communities (Fig. 3a). Results were similar in the 1997 experiment where, after 4 years growth, biomass increased with species richness ($F_{1,59} = 12.66$, $P < 0.001$) from 78.5 g m$^{-2}$ in monocultures to 183.4 g m$^{-2}$ in 16-species communities (Fig. 2). This 104.8 g m$^{-2}$ difference represented a 133% increase in biomass production. Complementarity effects explained 84.5 g m$^{-2}$ overyielding and selection effects explained 10.5 g m$^{-2}$ overyielding (Fig. 3b). Summarizing these two experiments, 16 species produced 118 g m$^{-2}$ more than monoculture plots due to 129 g m$^{-2}$ complementarity and −30.3 g m$^{-2}$ selection effects.

**Model predictions**. Plant-community simulation models that included a different growth rate for each soil training type (i.e. PSF simulation models) predicted that biomass would increase with species richness ($F_{1,59} = 7.81$, $P = 0.007$), from 60.1 g m$^{-2}$ in monocultures to 76.1 g m$^{-2}$ in 16-species communities (Fig. 2).

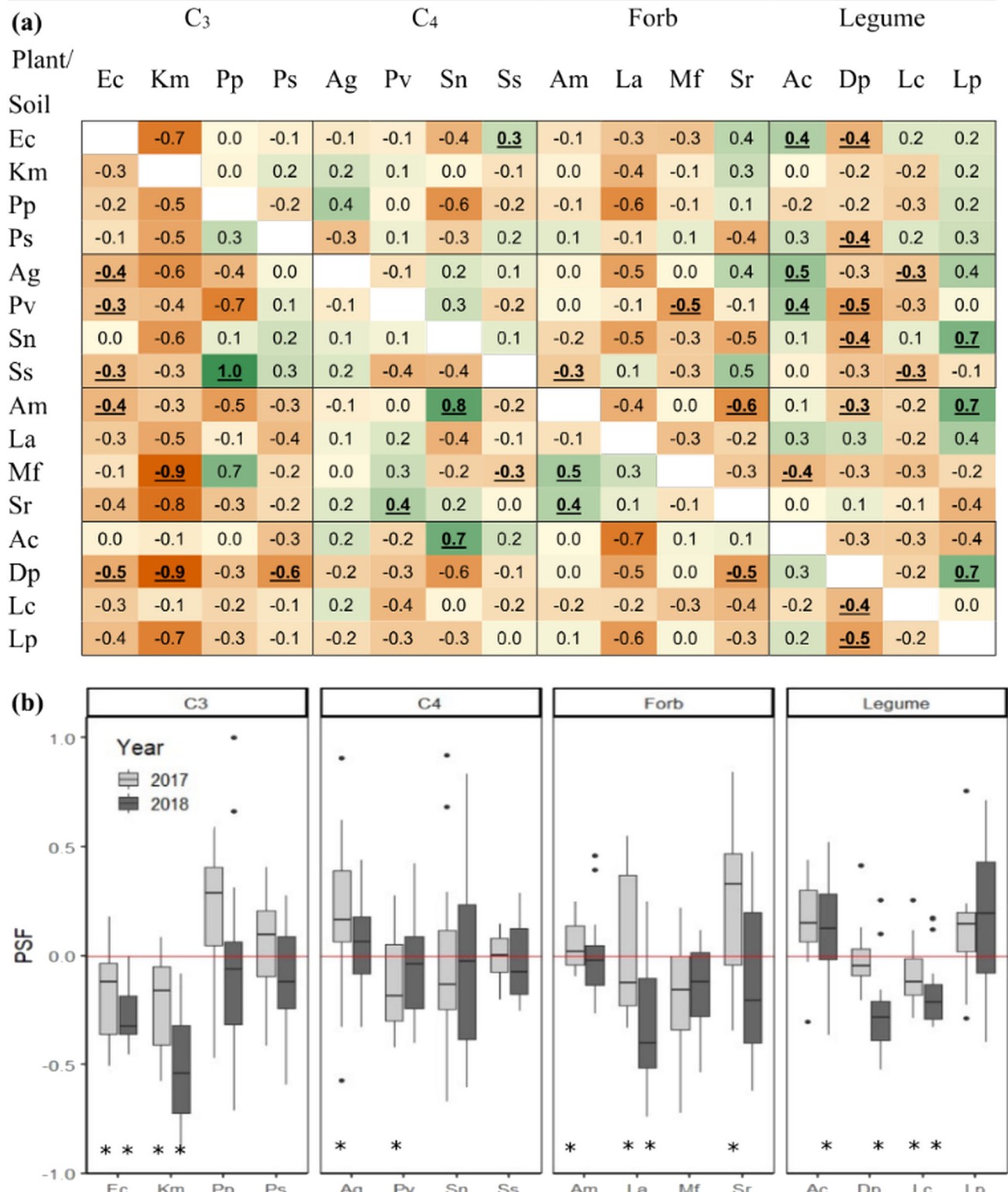

**Fig. 1 Plant-soil feedbacks (PSF) for 16 plant species.** PSF values from 2018 for each plant, on each soil-cultivation type shown in panel **a**. Orange and green highlighted values indicate negative and positive PSFs, respectively, and values with confidence intervals that do not overlap zero are bolded and underlined. Sample sizes derived from 27–35 replicates on 'self' soils and 5–9 replicates on each 'other' soil for each species. Exact sample sizes reported in Supplementary Table 1. Averaging across the 15 soil-cultivation type PSFs, produced one 'species-level' PSF value for each plant species (**b**). These species-level PSF values are shown for data from 2017 (grey) and 2018 (black). Each value represents the mean and standard error associated with the 15 soil-cultivation-specific PSF values. Asterisks indicate values that differed from zero in a one-sample $t$-test at ($\alpha = 0.05$).

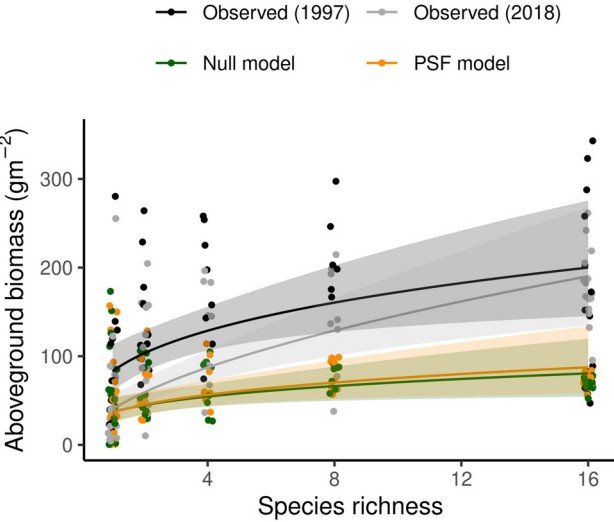

**Fig. 2 Observed and predicted plant biomass in experimental plant communities with one to 16 plant species.** Plant-community growth simulation models either with (PSF) or without (Null) plant-soil feedback effects predicted that biomass would increase with species richness (i.e. overyield). However, PSF simulation models correctly predicted this effect was caused by complementarity effects and Null models incorrectly predicted this effect was caused by selection effects (Fig. 3). The overyielding predicted by PSF simulation models represented 14% of the overyielding observed in the two biodiversity-productivity experiments. Each point represents total aboveground biomass in one community type ($n = 55$ or $63$ for the 1997 and 2018 experiments, respectively). Large values from six outlier plots are not shown but were included in analyses. Lines represent the best-fit curves and shaded areas indicate the 95% confidence intervals. In each dataset, biomass increased with species richness ($P < 0.05$; see Results for details).

This $16.0\,\mathrm{g\,m^{-2}}$ difference represented a 27% increase in biomass production and 14% of the overyielding observed in the two biodiversity-productivity experiments. Complementarity effects explained $15.0\,\mathrm{g\,m^{-2}}$ overyielding. Selection effects explained $1.0\,\mathrm{g\,m^{-2}}$ overyielding (Fig. 3c).

Null simulation models predicted that biomass would increase with species richness ($F_{1,59} = 7.33$, $P = 0.009$) from $60.7\,\mathrm{g\,m^{-2}}$ in monocultures to $70.7\,\mathrm{g\,m^{-2}}$ in 16-species communities (Fig. 2). This overyielding was caused by selection effects ($11.8\,\mathrm{g\,m^{-2}}$) and not complementarity effects ($-1.5\,\mathrm{g\,m^{-2}}$; Fig. 3d).

Both PSF and Null simulation model predictions were correlated with community biomass in the 2014 biodiversity-productivity experiment, though PSF simulation model predictions were closer to 1:1 (Observed biomass = $0.97 \times$ PSF predicted biomass + 47.3) and had a stronger predictive ability ($R^2 = 0.20$, $P < 0.001$, RMSE = 77.6) than Null simulation model predictions (Observed biomass = $0.81 \times$ Null predicted biomass + 61.5, $R^2 = 0.14$, $P = 0.003$, RMSE = 81.6). Removing two outlier communities with large biomass further improved correlations and resulted in $R^2$ values of 0.24 and 0.16 for PSF and Null simulation model predictions, respectively (Supplementary Fig. 1). Neither PSF nor Null simulation model predictions were correlated with biomasses from the 1997 biodiversity-productivity experiment ($P > 0.05$; Supplementary Fig. 1).

## Discussion

PSFs improved understanding of the magnitude and mechanism of the biodiversity-productivity relationship. In experimental communities, plants grew $118\,\mathrm{g\,m^{-2}}$ more in diverse communities than in monocultures. This occurred because most plants grew better than

expected from monocultures (i.e. complementarity effects) and not because dominant species were over-represented in communities (i.e. selection effects)[11]. PSFs helped explain this pattern because most plants created soils that decreased their own growth. Consequently, plants grew faster in communities, where they were surrounded by 'other' soils than in monocultures, where they were surrounded by 'self' soils[10,13,24]. This increased complementarity effects. Further, in Null simulation models, plants that grew most in monoculture were predicted to be over-represented in communities due to competition (i.e. selection effects). Negative PSF decreased these selection effects because dominant plants encounter higher proportions of 'self' soils than subdominant plants. The net effect of these changes was that PSF simulation models predicted $16.0\,\mathrm{g\,m^{-2}}$ more biomass in diverse communities than monocultures, due to complementarity effects. This $16.0\,\mathrm{g\,m^{-2}}$ represented 14% of the $118\,\mathrm{g\,m^{-2}}$ overyielding observed in experimental communities. PSF effects increased from 2017 to 2018 suggesting that PSF effects are likely to increase over time, though it is unlikely that PSFs would become a dominant determinant of overyielding. While 14% is a small portion of observed overyielding, results are important because they demonstrate diversity can increase productivity by suppressing plant disease. Results are also important because they help constrain the importance of other factors such as niche partitioning, which remain difficult to quantify[23].

The magnitude and direction of PSFs in this study were broadly consistent with those from across the literature[25,30,32,36], suggesting that PSFs likely play a similar role in other systems. The absolute value of PSFs (0.27) indicated that 2 years of plant growth created soils that changed subsequent plant growth by 27%. However, because PSFs were both positive and negative, the net PSF effect was smaller (i.e., a PSF value of $-0.10$ in 2018). Absolute PSF values reported across the literature tend to be larger (0.53)[36], but are mostly measured in greenhouse conditions that are known to exaggerate PSF values[27,36,37]. This research and previous modeling efforts suggest a direct negative relationship between PSF and overyielding[10]. In other words, PSFs that decrease plant growth by 10% on 'self' soils are expected to produce 10% overyielding. Because PSFs often change plant growth by 10–50% and overyielding often changes plant growth by 100–200%, we expect that PSFs will often explain 5–50% of overyielding[2,10,32,36].

PSF experiments are often performed by comparing plant growth on 'self-trained' soils to plant growth on soil trained by a non-specific mix of 'other' species[29,31]. At this species, or 'mixed-other' level, six species in 2018 realized significant PSFs. This 'mixed-other' approach has been criticized for overestimating PSF effects[31]. Our large factorial experiment allowed us to examine both 'mixed-other' and species*soil-level PSFs[31,38]. At the species*soil-level, 14 of 16 plant species realized a significant PSF on at least one soil type. While most species realized either positive or negative PSFs, three plant species demonstrated significantly positive PSFs on one soil type and significantly negative PSFs on a different soil type. For example, *A. canescens* PSF values ranged from $-0.4$ to $+0.5$. For this species, 'self vs. other' PSF experiments could be expected to report strongly positive, strongly negative or neutral PSF depending on the soil types used[31]. Variations in PSF, from positive to negative, were important for improving predictions of plant growth in communities. For example, positive PSFs helped improve correlations between predicted and observed community biomass by correctly decreasing the biomass of species with positive PSFs in communities (Supplementary Fig. 1)[10]. For example, a positive PSF for *L. perennis* on *S. nutans* soil, correctly resulted in less *L. perennis* biomass in *L. perennis/S. nutans* bicultures than predicted by the Null model. It should not be surprising that PSF vary as a function of the plant that trained a soil, but use of the

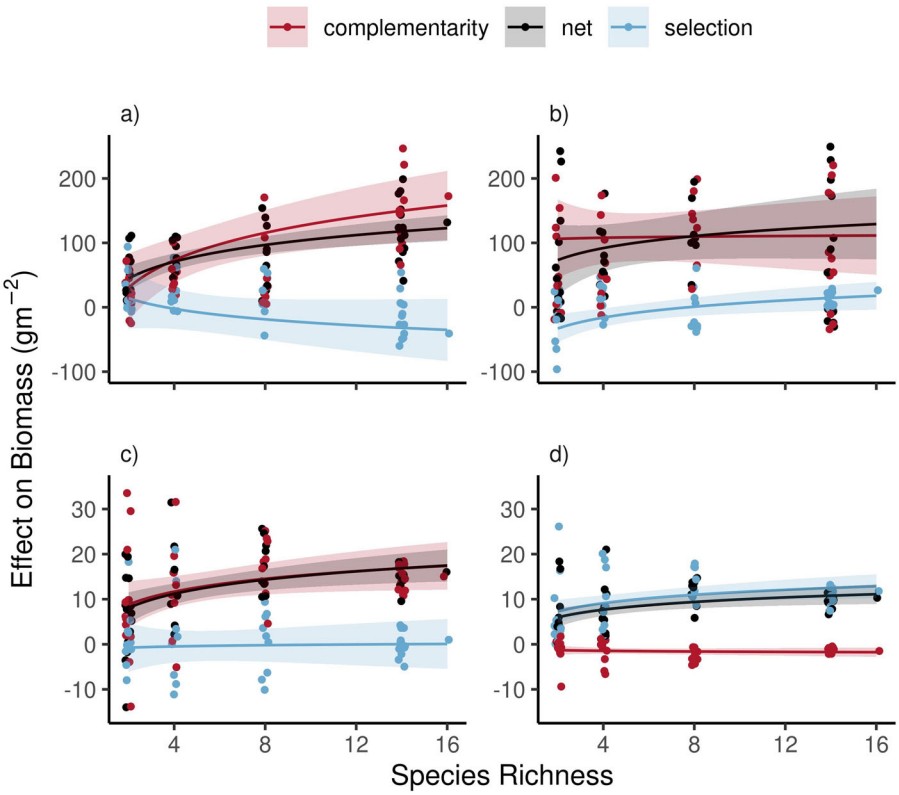

**Fig. 3 Observed and predicted plant-community biomass responses to species richness.** Data from experiments performed in 2014 and 1997 are shown in panels **a** and **b**, respectively. Data from simulation models that either included or excluded plant-soil feedback effects shown in panels **c** and **d**, respectively. Data from Net effects (black symbols) were separated mathematically into complementarity effects (red) and selection effects (blue). Plant-community simulation models that included plant-soil feedbacks correctly predicted that plant growth would increase with richness due to complementarity effects (**c**; red line) while the same models without plant-soil feedback effects incorrectly predicted positive selection effects (**d**; blue line) and negative complementarity effects (**d**; red line). Shaded areas indicate the 95% confidence intervals. Data from 63 (**a**) or 55 (**b**) different replicated plant communities. In each dataset, the net biodiversity effect increased with species richness ($P < 0.05$; see Results for details).

factorial designs needed to demonstrate this pattern remain rare[32]. Thus, results provide a clear example of how broad species-level assessments of PSF can hide important soil-type-specific PSFs[17,31,38].

PSFs in this and other studies tend to be negative, suggesting that plants accumulate species-specific pathogens. It is reasonable to expect that the negative effects of these pathogens would decrease with species relatedness, though evidence is mixed[34,35]. As the largest, factorial PSF experiment of which we are aware, this study provided a good opportunity to test for a phylogenetic effect, but we found no correlation between phylogenetic distance and PSF. This suggests that pathogen effects were highly species-specific and that phylogenetic distance beyond the population level may be inappropriate for generalizing PSF relationships[39].

In previous biodiversity-productivity experiments, species richness explained 18–46% of variation in biomass among communities[8,40,41]. In our 2014 experiment, richness explained 12% of the variation in community biomass, suggesting large variability among communities, likely due to smaller plots and a shorter experiment duration. Despite this variability, our Null plant-community model explained 12% of the variation in plant species biomass and our PSF model improved this correlation to 20%. Removing data from two large outlier communities improved these correlations to explain 16 and 24% of variation (Null and PSF simulation model predictions, respectively). Although correlations were not large, results demonstrate that it is possible to predict species biomass in communities with similar accuracy reported for higher levels of organization (i.e.

community biomass vs. community richness) that are generally assumed to be easier to describe[42–44].

While promising in the short-term, our models were not correlated with plant-community composition in the 1997 experiment. Because factors from climate to anthropogenic nitrogen deposition to soil microbial community composition have likely changed in the 20 years between these two experiments, it is impossible to pinpoint why community biomass differs between the two experiments[45]. An implication of the poor correlation between the new and old data is that inference about the effects of PSF on plant-community development are likely to be time- or site-dependent[46–49]. However, despite a lack of correlation between predicted and observed biomass for specific communities, the general pattern of increasing aboveground biomass with species richness was consistent across both experiments and PSF simulation model predictions.

## Conclusion

Biodiversity-productivity experiments were developed as a test of niche-partitioning effects, yet it remains difficult to quantify the extent to which niche partitioning determines biodiversity-productivity relationships[7,50,51]. Here, we report that PSFs explained 14% of the net biodiversity effect. Even though this effect is likely to increase over time, it is likely to remain modest relative to 100–200% increases in productivity across species richness treatments. Yet, demonstrating a 14% PSF effect is important because it quantifies how diversity can increase productivity in communities by

suppressing plant disease. It is also important because it helps constrain the role of other factors (i.e. niche partitioning) in biodiversity-productivity relationships[25,52,53]. Future research that quantifies and integrates niche partitioning with PSF and other effects can be expected to improve predictions of the effects of species loss on plant-community productivity and resilience with implications for biofuel production and conservation[6,54].

## Methods

Two-phase PSF experiments have become the standard approach to measuring PSF, though they are typically performed in greenhouse conditions using plant monocultures, and their effects on plant growth in communities are rarely tested explicitly[32]. Here, we used a two-phase field experiment to measure the growth rate of each of 16 plant species on soils trained by each of the 16 species in the experiment (i.e. a factorial PSF experiments). The plant growth rates from this experiment allowed us to simulate the growth of any combination of species. We used these growth rates to simulate the growth of 63 unique plant communities that were grown separately. We then compare model predictions to observed plant growth. To better understand the mechanisms determining biodiversity-productivity relationships, we used a standard mathematical approach to parse the net change in community biomass with species richness into complementarity and selection effect components[11].

Research was conducted in the Cedar Creek Ecosystem Science Reserve Long Term Ecological Research site, East Bethel, Minnesota, USA (45.403290 N, 93.187411 W). Previous research at the study site demonstrated large increases in community biomass with species richness (i.e. biodiversity-productivity relationships) that increase over time and are caused by complementarity[55]. Soils are sandy and of the Nymore series: mixed, frigid, Typic Udipsamment. During the four years of the study, mean annual precipitation and temperature were 723.0 mm and 6.5 °C, which is consistent with the 1963–2019 records at the site (769.3 mm and 6.6 °C, respectively).

We performed two experiments: a PSF experiment and a biodiversity-productivity experiment. Each experiment included 16 species used in an existing biodiversity-productivity experiment at the site (the Biodiversity II experiment; Table 1)[40]. Five species that together represented less than 3% of the biomass in the biodiversity-productivity experiment from 1997 (henceforth, BP$_{1997}$) were excluded from our PSF and biodiversity-productivity experiments due to seed availability (*Asclepias tuberosa* L., *Dalea villosa* Nutt., *Dalea candida* Michx) and poor growth in previous experiments (*Quercus macrocarpa* Michx., *Quercus ellipsoidalis* E. J. Hill). Seeds were purchased from Prairie Moon Nursery (Minnesota, USA), Granite Seed (Utah, USA), Prairie Restorations Inc. (Minnesota, USA) and Minnesota Native Landscapes (Minnesota, USA).

In October 2014, a 1750-m² fallow area adjacent to the BP$_{1997}$ experiment was sprayed with a 5% glyphosate solution (Monsanto, Missouri, USA) and disc-harrowed to 15 cm to incorporate vegetation and homogenize soils. For the PSF experiment, 2720 plots (0.75 × 0.35 m) were established. For the biodiversity-productivity experiment, 232 plots (1.5 m by 1.5 m) were established. Plots were immediately adjacent to one another, but for all plots, a 35-cm deep by 4-cm wide trench was dug and lined with a root barrier to ensure that plant roots grew in target soil conditions (1-mm thick high-density polyethylene; Global Plastic

**Table 1 Plant species and functional groups used in the plant-soil feedback and biodiversity-productivity experiments.**

| Species | Functional group | Code |
|---|---|---|
| Amorpha canescens | Legume | Ac |
| Andropogon gerardii | C$_4$ | Ag |
| Achillea millefolium | Forb | Am |
| Dalea purpurea | Legume | Dp |
| Elymus canadensis | C$_3$ | Ec |
| Koeleria macrantha | C$_3$ | Km |
| Liatris aspera | Forb | La |
| Lespedeza capitata | Legume | Lc |
| Lupinus perennis | Legume | Lp |
| Monarda fistulosa | Forb | Mf |
| Poa pratensis | C$_3$ | Pp |
| Pascopyrum smithii | C$_3$ | Ps |
| Panicum virgatum | C$_4$ | Pv |
| Sorghastrum nutans | C$_4$ | Sn |
| Solidago rigida | Forb | Sr |
| Schizachyrium scoparium | C$_4$ | Ss |

Sheeting, California, USA). Throughout the PSF and biodiversity-productivity experiments, non-target plants were removed by hand several times each year.

**PSF experiment**. A two-phase, factorial PSF experiment was used[28,29]. Phase I began in April 2015. For each of the 16 target species, 10 g live seed m$^{-2}$ was planted by hand in 170 replicate plots. During 2015, plots were watered weekly to promote establishment, and during the first 2 years plots were weeded once every 2 weeks to ensure the conditioned soils were monospecific. Seeded plant species grew in 2608 of the 2720 plots in Phase I. It is critical that Phase I plants do not re-establish from roots in Phase II because this growth would appear as a positive PSF. To prevent re-sprouting, vegetation was killed with a 5% glyphosate treatment and aboveground biomass removed, plots were then hand-tilled with a garden claw (~75% of plots) or rototiller as necessary (~25% of plots; Stihl Inc., Delaware, USA), November 2016. Finally, plots were again treated with herbicide in April 2017, prior to Phase II seeding, which replicated Phase I seeding rates. Glyphosate application may affect mycorrhization and therefore decrease positive PSF[56], but it was critical to ensure that all Phase I plants were killed because re-sprouting plants have the potential to create large, false positive PSFs.

For Phase II, each target species was to be planted in 35 replicate plots with 'self' soils and nine replicated plots with each of the 15 'other' soils. Because some target species failed to establish in Phase I, actual replication ranged from 27 to 35 replicates on 'self' soils and five to nine replicates on each 'other' soil (Supplementary Table 1). Further, each target species was randomly assigned to five to nine replicate plots that had no Phase I growth. These 'control' plots had no plant growth during Phase I and were used to parameterize one of the Null models. During Phase II, plots were weeded once per month.

Plant cover in every plot was assessed by visual estimation in August 2017 and September 2018 and plant aboveground biomass was clipped, dried and weighed in October 2018. The 2017 percent cover data was converted to biomass values using the 2018 percent cover to biomass relationship.

**Biodiversity-productivity experiment**. In April 2015, 63 plant communities containing 1–16 plant species were planted in 232 plots. Plant communities with 1, 2, 4, 8, 16, and 16 species were established with 16, 14, 9, 9, 14, and 1 unique community compositions for each richness level, respectively (Supplementary Data 1). Each unique community composition was planted in three replicate plots, except monocultures which were each planted in four replicates plots, and 16-species communities which were planted in 30 replicate plots. Community compositions were designed to replicate those in the BP$_{1997}$ experiment (Supplementary Data 1)[55,57]. For 40 of 63 communities, species composition in the new and existing experiments were identical. The remaining 23 communities differed in that they did not include the five species described above, but again, these species represent less than 3% total biomass in BP$_{1997}$.

Each plot received 10 g live seed m$^{-2}$, with each seeded species in the community representing equal proportions of the seed mix. Plots were watered in the first year of the study (2015) and were weeded every two weeks for the first 2 years of the study. Thereafter, plots were weeded once per month. In August 2017, percent plant cover by species was assessed by visual estimation to the nearest percent. Rather than removing thatch by burning (as in BP$_{1997}$), total biomass was harvested and removed to prevent melting the plastic root barrier. In August 2018, plant cover in each plot was assessed by visual estimation, then randomly selected 15 cm by 150 cm strips were clipped, sorted to species, dried to constant weight at 60 °C and weighed to the nearest 0.1 g. The remaining biomass was then clipped, dried and weighed. Percent cover to dry biomass correlations were used to transform percent cover values to biomass values.

To provide an additional test of the role of PSF in the BP relationship, we also used published data from the fourth year of the BP$_{1997}$ experiment (https://www.cedarcreek.umn.edu/research/data). Cover to biomass relationships reported for 2007 were used to convert species-level cover data to species-level biomass that were then scaled to match observed community biomass[40].

### Statistics and reproducibility

*Calculating and testing PSF values*. PSF index values were calculated from above-ground biomass data as follows: PSF = (S-O)/max(S,O) where S is the aboveground biomass produced in Phase II on 'self' soils, O is the aboveground biomass produced in Phase II on 'other' soils, and max(S,O) selects the larger of S and O. This calculation and the commonly used log response ratio have been found to be superior to other calculations, but the calculation we use has the added benefit that it produces values that are bound by −1 and 1 and are easily interpretable as the proportion change in growth among soil types[29]. The mean and error associated with these values was estimated using bootstrapped confidence intervals calculated using the sample_n command from the R package 'dplyr'. Because PSFs were measured for 16 species on 15 soil types, analyses yielded 240 species*soil-level PSF values. Because the mean of large positive and large negative PSF values can be zero, and therefore 'mask' PSF effects, we also calculated the absolute value of PSF values.

The 240 species*soil-level PSF values were considered positive or negative when their 95% confidence interval did not overlap zero. Variation in species*soil PSF values is derived from the 27–35 replicate "self" and 5–9 replicate "other" field plots. Species-

level PSF values were then calculated as the mean PSF value across 15 soil training types. Variation in species-level PSF is derived from the 15 soil types. To determine if species-level PSF values differed from zero, one-way t-tests were used. Species-level PSF were considered different from zero when $P < 0.05$. To test whether or not PSF values changed between the first and second year of Phase II, a one-way ANOVA with year as a factor was used ('aov' and 'TukeyHSD' in R programming). Differences among years were considered significant when $P < 0.05$. Differences among functional groups were tested with a t-test and effects of phylogenetic distance were tested using a correlation between phylogenetic distance and species*soil-level PSF values[58].

*Plant-community growth simulation models.* PSF experiments describe plant growth on soils trained by different species, but do not describe how plants grow in communities. To assess how these PSFs are likely to affect plant growth in communities, we use plant-community simulation models with and without PSF effects to predict plant biomass and we compare model predictions to plant biomass observed in experimental plant communities. Broadly, these models allow each plant in a community to grow from seed at rates determined from the PSF experiment. Plant growth is eventually limited by a carrying capacity. The best-performing discrete plant-community simulation models in a similar previous study were used (i.e. the 'logistic species-level-K model' and the 'logistic constant-K model')[59,60]. In this logistic growth simulation model, species-conditioned soils 'grow' as a function of plant biomass, plant species growth rates, and a plant-to-microbe conversion factor. Plant growth rates are a function of the proportion of different soil training types present. To prevent run-away growth, biomass is limited by a carrying capacity, which can be either unique to a species or to the community. Null model simulations are the same except that they include only one soil training type and one plant growth rate (Supplementary Note 1). The Null version of these models does not include a complementarity mechanism, but they can produce selection effects.

Growth rates were derived from (a) growth on control soils (control Null model), (b) growth on 'self' soils (self Null model), or (c) growth on each soil type (PSF model). Competition coefficients were assigned a value of '1', but each species could affect the growth of other species due to community-level carrying capacities[60]. Each of these three model parameterizations (i.e. growth on control, growth on self, or growth on each soil type) was run with five different carrying capacities: (1) the maximum observed growth in any plot in the community experiment, (2) the maximum mean observed growth in any community, (3) the maximum species-specific growth in community plots, (4) the maximum observed growth in any PSF plot, and (5) the maximum species-specific growth in any PSF plot. Mean Null model predictions of community biomass were calculated from the 10 model simulations (Control Null, Self Null each with five carrying capacities). Mean PSF-model predictions were calculated from the five simulations with different carrying capacities.

Because growth rates were derived from the second year of growth, we assumed that growth rates represented 2 years of growth. To simulate the four years of growth in the biodiversity-productivity experiment, model simulations were executed for 52 timesteps, after which plant biomass was reduced to 1% of the previous timestep and allowed to run for another 52 timesteps. Model simulations for 52 or 208 timesteps produced qualitatively similar results but only results from the 104 timestep approach described immediately above are reported since they best represented conditions in the field. Mean model output for the sum of species growth from the suite of Null or PSF-model simulations are reported.

*Parsing selection and complementarity effects.* To better explain why biomass changes with species richness in each dataset (observed or predicted), the net change in community biomass with species richness was parsed into complementarity and selection effect components, using the modified Price equation (R package 'partitionBEFsp')[11,33]. Complementarity effects can be positive or negative, depending on whether species on average have higher or lower yields than the expected relative yield. Selection effects can be either positive or negative, depending on whether species have a positive or negative covariance between relative yield and biomass. This method is easily interpretable, comparable to other results, and remains the standard practice. Data from outlier communities with total biodiversity effects greater than five times the interquartile range were removed. Because *S. rigida*, *D. purpurea*, *D. villosa*, and *D. candida* did not grow in monoculture communities in BP$_{1997}$, when partitioning biodiversity effects for BP$_{1997}$, monoculture growth for these species was assumed to be twice biculture growth.

*Testing PSF and biodiversity-productivity data.* Patterns in the observed and predicted biomass with species richness were described with simple, best-fit log linear regressions (Proc Reg; SAS V9.4). The relationship between predicted and observed biomass in different plant communities was assessed by ordinary least squares regression. Plant-community biomass was the response variable that was predicted by either Null-or PSF-model-predicted biomass.

**Reporting summary**. Further information on research design is available in the Nature Research Reporting Summary linked to this article.

## Data availability
All data used to prepare this manuscript can be found deposited at USU Digital Commons: https://doi.org/10.26078/52k0-jr94.

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

## Acknowledgements

This work was supported by grants from the US National Science Foundation DEB-1354129, and the Utah State University Ecology Center. This research was supported by the Utah Agricultural Experiment Station, Utah State University, and approved as journal paper number 9432. In addition, this work incorporated data from the Biodiversity II experiment at Cedar Creek Ecosystem Science Reserve, which is supported by grants from the US National Science Foundation Long-Term Ecological Research Program (LTER) including DEB-0620652 and DEB-1234162. Thanks to T. Mielke, K. Worm, J. Krueger, M. Saxhaug, J. Anderson, P. Barnes, L, Broome, J. Suvada, M. Berndt, A. Lindsey, J. Borchardt, B. Terry, J. Allenbrand, C. Pint, C. Carlisle, A. Brooks, A. Yamaguchi, A. Zlevor, L. Cherubini, P. Guevarra, L. Korte, M. Koenig, and C. Johnson for assistance with the field experiment. S. Durham and M. Holdrege provided statistical assistance. P. Adler, K. Beard, L. Kinkel, and S. Kuehl-Shelby provided manuscript feedback.

## Author contributions

L.F., A.K., J.G., and J.N. planned and designed the experiment. L.F., A.K., and J.G. performed the experiment. L.F. collected data. L.F. and A.K. collaborated in interpreting data. All authors contributed to the generation of ideas in this manuscript.

## Competing interests

The authors declare no competing interests.
