## [Peer Review File · Communications Biology]

Reviewers' comments:

Reviewer #2 (Remarks to the Author):

The authors describe results from three field experiments (one PSF experiment and two biodiversity experiments) as well as modeling that either included PSF effects or not. The authors found that the PSF effects explained 12-15% of the overyielding in biodiversity experiments, and that including PSF responses in models better predicted the underlying mechanisms (complementarity vs. selection effects).

I really like the fact that the authors did a PSF experiment in the field containing an unprecedented number of species and thousands of plots (albeit very small field plots), as this is where we need to do experiments for increased realism. It's also nice that surveys allowed for estimates of changes in PSF over time.

I am not a modeler and can therefore not comment on that aspect of the study although I can point out what is confusing to me with the hope that it may help the authors clarify aspects that may also confuse others. First, it is unclear to me what the authors mean when they say PSF models. Does the model predict the plant community total growth or the changes in growth across a plant diversity gradient that is supposedly due to PSF? If it's the former, it's doing a pretty poor job judging from the abstract and Figure 2 and severely underpredicts productivity. Also, it's not looking all that different from the null model without PSF in Fig. 2 and is this where the authors estimate the 12-15% contribution of PSF to the overyielding?

Other than that, I found it relatively easy to follow, but I wonder if the Discussion would be clearer by organizing it into subheadings making some main points?

Below are some more specific comment that I hope are useful when the authors revise the current manuscript.

- Abstract. The third bullet point is confusing without the proper background. For example, while the PSF model is better as it predicts more overyielding due to complementarity, what do we compare that to so that we understand that the model without PSF incorrectly assign the overyielding to selection effects? Both seem pretty bad if we are comparing that to experimental findings where complementarity explained 185% of the overyielding. Also, under bullet point four, I wouldn't say that the PSF model improved the magnitude of predictions as it only went from 17 to 27% whereas the real (?) value was 185%. Am I missing something here or do both models severely underpredict overyielding? Or, does the PSF model predict the proportion explained by PSF, but if that is the case, I do not understand why the PSF models predict complementarity (lines 315-317).
- Line 37. Correct spelling of overyielding.
- Lines 40-41. I think you mean mutualists, not symbionts as "symbiont" just mean living in close proximity but says nothing about the nature of the interaction. Also, I can see species-specific mutualists with legumes, but given that this is a grassland community, most plants will associate with AMF that are not particularly specific so I see no obvious reason why they would be more abundant (or beneficial) in monocultures as it all depends on host quality.
- Lines 73-76. Shouldn't one factor also be PSF in your model?
- Line 102. What was the distance between plots?
- Lines 115-119. This fallow period is a lot longer than many greenhouse experiments use, which may be one reason for why significant PSF effects were only observed in 15% or so of the treatments? That may be worth commenting on in the Discussion?
- Line 121. Seeded? If so, by using the same seed density as in Phase 1?
- Line 135. Probably a good idea to also outline what the "maximum" S and O refer to.
- Lines 140-146. I find "soil type" to be distracting here as all experiments were conducted on one soil type. I would prefer soil training. Also, does the 5-9 other replicates mean for each other species the focal plant was grown on, making the total other reps (5-9) x 15?
- Lines 149-150. Given that you have 16 species and you ran t-tests on all of them, how do you control for false positive if the individual error rate is 0.05? Seems like at least one could be significant just by chance if not controlled for. Then again, if you do control for multiple tests it

becomes so conservative as to become meaningless so not sure what the best approach is here.

- Lines 247-250. Does this mean that only 15% (36/240) actually showed a significant PSF and that the great majority were neutral? Also, 13+23 is 36, not 39 unless I am missing something here.
- Line 269. How can you have selection effects in a community that contain all the 16 species given that there is zero change that a particularly productive species wouldn't be selected? I must be missing something here.
- Lines 303-304. The fact that PSF explain 12-15% of overyielding in biodiversity experiments seems "abstract-worthy". Also, where does that estimate come from? It's not anywhere in the Results so is it the difference between the null model and PSF model?
- Lines 328-330. I do not understand why the PSF model would decrease the biomass of positive PSF. If plants experience positive PSF, shouldn't the model promote their biomass relative to a null community without PSF? What am I missing here?
- Lines 334-336. Do the other studies that are reference also report absolute PSF? I am more familiar with PSF means.
- Fig. 1a. It would be good to use vertical lines to separate the functional groups. Also, could the authors do some sort of analysis to see if there is any relationship between phylogenetic relatedness and PSF? For example, was PSF more negative when grown after a more closely related species as would be predicted if pathogens are species specific and respond to similarities in traits (if traits are phylogenetically conserved that is)? It could help us get to underlying mechanisms to better understand and predict PSF. Could the authors correlate pairwise phylogenetic distance among plants with PSF?
- Fig 2. I think what I am having a hard time understanding is whether the plotted PSF line is the predicted contribution of PSF to the model that incorporate PSF or if the red line represents the predicted total biomass from the whole model that includes PSF. If the former, do you get the 12-15% contribution by PSF from the difference between the red and green line? I hope this confusion help the authors clarify things that may be crystal clear to them but not to a non-modeler like myself....
- Fig. S1. I don't see any blue here.

Reviewer #3 (Remarks to the Author):

This manuscript describes results from a group of field experiment that estimated how much plant-soil feedbacks contributed to a positive diversity-productivity relationship in a grassland community. Overall, plant soil feedbacks (performances of plant species in own versus other soil) were negative and varied considerably for each plant/soil combination. The diversity productivity relationship was strongly positive and driven by complementarity effects. Plant community growth models that either included or did not include measured plant-soil feedbacks were used to predict total biomass in the experimental diversity communities. Plant-soil feedbacks predicted about 15% of the biodiversity-productivity relationship and predicted complementarity as the predominant mechanism. However, the models did not predict biomass in a longer-term plant-soil feedback experiment.

First, I'd like to commend the authors on pulling off an ambitious, well-designed experiment that contained almost 3,000 (amazing!!) plots that were maintained for four years. While several hundred diversity-productivity experiment have been conducted, the mechanisms underlying diversity-productivity relationships have remained elusive. Plant-soil feedbacks certainly offer one potential mechanism that can be more readily tested than other mechanisms, such as niche partitioning. While a handful of other experiments have linked plant-soil feedbacks to diversity-productivity relationships before, to my knowledge this is the first two-phase field experiment – which likely increases realism relative to greenhouse experiments. It is also very well-replicated (a common failing in plant-soil feedback experiments), increasing confidence in experimental results.

I have a few comments that may improve the manuscript:

1. It is likely that plant-soil feedbacks vary with diversity (e.g., Hawkes et al. 2013) and the frequency/density of conspecific individuals (e.g., Chung and Rudgers 2016). Measuring these

differences and incorporating them into the plant community growth models are certainly outside of the scope of this experiment, but I do think the issue should be addressed in the discussion. With more fine-scale information, plant-soil feedback may become a better (or worse) predictor of plant biomass in diverse communities.

Chung, Y. A., and J. A. Rudgers. 2016. Plant–soil feedbacks promote negative frequency dependence in the coexistence of two aridland grasses. *Proc. R. Soc. B* 283:20160608.

Hawkes, C. V., S. N. Kivlin, J. Du, and V. T. Eviner. 2013. The temporal development and additivity of plant-soil feedback in perennial grasses. *Plant and Soil* 369:141–150.

2. Glyphosate was used to remove unwanted plants when the experiment was established and after the conditioning phase of the plant-soil feedback experiment. Does glyphosate have any known effects on soil microbial communities? This may be a concern for measuring feedbacks if it disproportionately affects mutualists or pathogens – although effects are equal across the plant-soil feedback experiment. However, the glyphosate was not reapplied to the biodiversity experiment. If glyphosate influenced soil microbes, it may influence correlations between biomass predicted in plant-soil feedback models and the observed biomass in the diversity experiment.

3. In the results from the plant-soil feedback experiment, there is evidence that plant-soil feedbacks become more negative (on average) through time. If they continue to change, what effect does this have on model predictions of community biomass?

4. Null models did not predict the observed complementarity effect, but plant-soil feedback models did, suggesting that plant-soil feedback models do a better job explaining community biomass than null models, despite both predicting the positive diversity-productivity relationship. Can null models produce complementarity effects? It seems like the answer is no since there is no chance for another mechanism (niche partitioning) to produce complementarity effects in the plant-soil feedback experiment, but I may be missing something? Also, how do the models hold up to predicting plant community composition in diverse communities? Is one better than the other?

Minor edits:

Figure 2 caption: no need for “a” in “in a new”

Response to review comments

Thanks for this opportunity to revise our manuscript. As suggested in the ‘revision checklist’, we have numbered reviewer comments and indicate our responses with the same number and an ‘r’ (e.g., our response to reviewer comment 1 is listed immediately following comment 1 and indicated with a ‘1r’). As recommended by the editor, our revision ‘1) strengthens the data analysis, and 2) improves the presentation, discussion and interpretation’. These revisions have resulted in a thoroughly-revised, clearer manuscript.

More specifically

1. We strengthen data analysis by:
 - a. including a new phylogenetic analysis as suggested by reviewer 2.
 - b. Providing detailed support for our use of the $(S-O)/\max(S,O)$ calculation instead of the $\ln(S/O)$ calculation of PSFs. We have taken several steps to clarify our modelling approaches.
2. We improve the description of our data analysis
 - a. We provide a new summary of our approach at the end of the Introduction.
 - b. We also refer to the community growth models as ‘community growth simulation models’ throughout the paper. We expect that this will help distinguish model predictions from the calculations that parse net biodiversity effects into selection and complementarity components.
3. We improve presentation, discussion and interpretation by addressing reviewer comments.
 - a. Reviewers noted that PSFs explained a modest portion of overyielding and that correlations between predicted and observed values were low. We now address these concerns directly. In doing so, we have improved clarity and better place our results in context. Specifically, we note that few studies have attempted to predict biomass of specific plant community compositions, due to inherently large variation, so our efforts to do this represent an important advance. Similarly, as noted by the reviewers, it remains notoriously difficult to quantify the mechanisms driving the biodiversity-productivity effect, so our efforts at quantifying PSF effects represent an important advance in developing a quantitative understanding of the biodiversity-productivity relationship.

We expect that you will find that we have significantly improved the manuscript. Details are indicated in the point-by-point responses.

Reviewer 1.

1. The aim of this study is to examine the degree to which PSF mediate a positive diversity-productivity relationship. The authors conducted a massive PSF experiment in the field using 16 species growing in 2720 plots. Besides, a biodiversity-productivity experiment was also established containing 1 to 16 plant species. Plant community growth models were used to predict plant species biomass in communities, and combining with observed biomass to parse the complementarity and selection effects to explain biodiversity-productivity relationships, The

authors found the PSF effect in the second phase was 27% on affecting plant growth, and the model with PSF predicted overyielding mostly resulting from complementarity effect, while the model without PSF showed selection effects. Overall, this manuscript reflects a well-designed experiment, complemented by a suite of interesting modeling analyses, which strengthens current understanding and mechanistic acknowledgements on how PSF mediate biodiversity-productivity relationships.

1r. Thanks for this summary, we agree that this substantial experiment and analyses greatly advance our understanding of the role of PSF in plant communities.

General comments:

2. The authors used $(S-O)/\text{maximum}(S,O)$ to calculate the PSF effect and the PSF value was used to calculate community-level PSF, and overyielding effect. However, according to Brinkman et al. 2010, there are several ways in the PSF calculation, how does the authors think the current equation is the best to represent PSF effect and explain overyielding in the diverse plant community?

2r. This is an important point. Brinkman et al. (2010) conclude that the calculation we use, and the commonly used log response ratio calculation are similar and superior to other calculations because they both produce values that are symmetric around neutral PSF values (Brinkman et al. 2010, Fig. 3) and both have nearly identical ability to detect PSF effects (i.e., statistical power; Brinkman et al. 2010, Fig. 4).

The calculation we use has two advantages: it is readily interpretable as the proportion change in growth and, because it is bound by -1 and 1, it does not produce large values that are likely to overemphasize unusually large PSFs. We now note these points, in the methods (line 320-325).

To demonstrate this, we compared values derived from the two techniques here. There was no difference in a t-test ($P=0.09$). Only one value (for Koecr) was qualitatively different between the two values. This difference occurred because $\ln(\text{self/other})$ calculations produce large values (up to about 10) at the tail end of the distributions (very large positive or negative values) whereas our calculations are not as biased by these large 'tail' values.

Further, PSF value calculations are used in our paper to provide a general summary of PSF effects. This calculation was not used in our models, and so this calculation does not affect our results. We now note this at the beginning of the results (line 66). Our models use the raw plant growth data and not the calculated PSF values. To make this point clearer, we have moved the following sentence up to the beginning of this section (line 66): ‘The PSF experiment was performed, primarily, to produce plant growth rates on different soil training types to be used in plant community simulation models, but we also report PSF index values because they are a common metric that provide a simple summary of plant-soil interactions²⁹’

Here we have plotted the PSF values from this study using our $(S-O)/\max(S,O)$ values (x axis) versus $\ln(S/O)$ values (y-axis). Our calculations (x-axis values) have two benefits. First, they are readily interpretable as the proportion of change in growth. Second, because our values are bound by -1 and 1, our values are less susceptible to rare, large values. For example, the four

very large values calculated by the $\ln(S/O)$ approach would unreasonably skew mean and error values. Again, the critical point here, which we now clarify through the manuscript, is that the PSF value is not used in our models, so the calculation used does not affect our results, rather it is just a way to summarize PSF effects.

3. Why the modeling approach was used since the authors already did a manipulative field PSF experiment? It seems that the growth rate was used to simulate four years of plants growth, but based on the previous content, both second-phase feedback experiment and the B-P experiment were harvested in 2017 and 2018, wasn't it matched between each other and why need four years data? By the way, the observed and predicted biomass data seems didn't show very strong correlation between each other (Fig. S1). Please add some information to be clearer.

3r. The two experiments ran for different lengths of time. In the B-P experiment, plants grew for four years. In the PSF experiment, plants were grown for two years to create soil cultivation types, killed, then we observed the growth of each species on each soil type for two more years, so in 2018 plants in the B-P experiment had been growing continuously for 4 years and plants in the PSF experiment had been growing for 2 years continuously.

Measuring 240 PSF values in a factorial design provides the data needed to describe millions of potential plant community interactions (16! Combinations). The alternative would be to perform PSF experiments for whole plant communities. This is an excellent idea, and one that has been performed by Kulmatiski (2018), but has the short coming that every plant community would require it's own PSF experiment. For example, the 240 psf values we produced could have described the growth of 16 plant communities on soils cultivated by 15 different plant communities.

To clarify this point, we have added the following to the methods (at line 226): 'Two-phase PSF experiments have become the standard approach to measuring PSF, though they are typically performed in greenhouse conditions using plant monocultures, and their effects on plant growth in communities are rarely tested explicitly. Here, our general approach was to measure PSFs in the field. PSFs were measured using a two-phase approach to describe all the potential PSFs for each of 16 plant species. This data allows us to simulate the growth of any combination of species. We use this data to simulate the growth of 63 different plant communities that were grown separately and explicitly compare model predictions to observed plant growth.'

In regards to the correlation between observed and predicted community biomass, we now address this directly in the discussion and we feel that doing so has strengthened the paper. In short, our predictive ability is comparable to other studies. More specifically, we have added the following to the Discussion: 'In previous biodiversity-productivity experiments, species richness has explained 18% to 46% of variation biomass among communities (Tilman et al. 1996; Hector et al. 1999; Fornara et al. 2009). In our 2014 experiment, richness explained 12% of the variability in community biomass, suggesting large variability among communities, likely due to smaller plots and a shorter experiment duration. Despite this variability, our Null plant community model explained 12% of the variation in plant species biomass and our PSF model improved this correlation to 20%. Though correlations were not large (*i.e.*, >50%), results demonstrate that it is possible to predict species biomass in communities with similar accuracy reported for higher levels of organization (*i.e.*, community biomass vs. community richness) that are generally assumed to be easier to describe (Laughlin et al. 2017; Metcalfe et al. 2020; Moulin et al. 2021).'

Fornara, D.A. and Tilman, D., 2009. Ecological mechanisms associated with the positive diversity–productivity relationship in an N-limited grassland. *Ecology*, 90(2), pp.408-418.

Hector, A., Schmid, B., Beierkuhnlein, C., Caldeira, M.C., Diemer, M., Dimitrakopoulos, P.G., Finn, J.A., Freitas, H., Giller, P.S., Good, J. and Harris, R., 1999. Plant diversity and productivity experiments in European grasslands. *science*, 286(5442), pp.1123-1127.

Laughlin, D.C., Strahan, R.T., Moore, M.M., Fulé, P.Z., Huffman, D.W. and Covington, W.W., 2017. The hierarchy of predictability in ecological restoration: are vegetation structure and functional diversity more predictable than community composition?. *Journal of Applied Ecology*, 54(4), pp.1058-1069.

4. I like figure 3, but I think more details should be provided in M&M part on how the PSF or null model effect on biomass was calculated (fig. 3c,d). It seems the data was predicted from the modelling, however, it's unlikely to be true incorporating this simulated data into real overyielding effect observed in the field and drew a conclusion that PSF contribute 12 to 15% overyielding (Line 304).

4r. Thanks for this comment. Several reviewer comments have addressed difficulty in understanding the modelling approach and application, so we have tried to make our approach much clearer throughout the paper. Basically, 1) we observed plant growth in communities in the field. 2) we used a community growth simulation model to simulate how each of the species grow together. This model was used to simulate plant growth with one growth rate for each species (Null model) or with different growth rates on cultivated soil types. 3) For these three datasets (observed, Null simulation, PSF simulation), a mathematical approach was used to separate net overyielding into selection and complementarity components.

We now include this new summary at the end of the Introduction. We now refer to the community growth models as ‘community growth simulation models’ to help clarify what they do and how they are different from the mathematical partitioning of net overyielding into selection and complementarity components.

More specifically, we now state (lines 52-61): ‘To do this, we grew 16 species on soils cultivated by each species in the experiment (i.e., a factorial PSF experiment). We used these soil-specific growth rates in a suite of plant community growth simulation models and compared model predictions to plant growth observed in experimental communities. We also ran the model with one growth rate for each plant species (i.e., a Null simulation model with no PSF effects) to test how PSFs effected the model. To better understand the mechanisms determining how community biomass changes across species richness levels, we separated net biodiversity effects in each dataset (observed, Null, PSF) into complementarity and selection effect components (Loreau & Hector, 2001; Clark *et al.*, 2019). Because this experiment produced 240 PSF values, it was also possible to test if PSF decreased with phylogenetic distance which, if found, would help generalize PSF effects in the biodiversity-productivity relationship (Anacker *et al.* 2014; Mehrabi and Tuck 2015).’

We have also rewritten the Fig. 3 legend as follows to clarify our approach: ‘Observed (a and b) and predicted (c and d) biomass responses to species richness. Net effects (black symbols) were separated mathematically into complementarity effects (red) and selection effects (blue). Plant

community simulation models that included plant-soil feedbacks correctly predicted that plant growth would increase with richness due to complementarity effects (c; red line) while the same models without plant-soil feedback effects incorrectly predicted positive selection effects (d; blue line) and negative complementarity effects (d; red line).'

We have also added the following at the beginning of the Materials and Methods to clarify our approach (line 226): 'Two-phase PSF experiments have become the standard approach to measuring PSF, though they are typically performed in greenhouse conditions using plant monocultures, and their effects on plant growth in communities are rarely tested explicitly (Crawford et al. 2019). Here, we used a two-phase field experiment to measure the growth rate of each of 16 plant species on soils cultivated by each of the 16 species in the experiment (i.e., a factorial PSF experiments). This data allows us to simulate the growth of any combination of species. We used plant growth rates on soils cultivated by different plants to simulate the growth of 63 unique plant communities that were grown separately. We then compare model predictions to observed plant growth.'

We have also added the following to the Methods section (line 347): 'PSF experiments describe plant growth on soils cultivated by different species, but do not describe how plants grow in communities. To assess how these PSFs are likely to affect plant growth in communities, we use plant community simulation models with and without PSF effects to predict plant biomass and we compare model predictions to plant biomass observed in experimental plant communities. Broadly, these models allow each plant in a community to grow from seed at rates determined from the PSF experiment. Plant growth is eventually limited by a carrying capacity.'

Line edits:

5. Line 91 need a space before the bracket.

5r. Thanks, we have fixed this.

6. Line 104 what's the purpose of the polyethylene root barrier, to separate the plot?

6r. We now note that root barrier was used to ensure that roots grow in target soil types (line 262).

7. Line 119 should be April 2017?

Line 124 should be Table S2

7r. November 2016 is correct. As indicated, plots were herbicided in late summer, harvested, then the plots were tilled in November 2016. Plots were then treated with herbicide again in April 2017 to ensure that all Phase I plants were killed. We have rephrased these sentences to clarify this point (line 276).

8. Line 125 what's the treatment or what kinds of plants growing in for the control plot?

8r. These plots were maintained free of vegetation during Phase I. We have added a note to this effect on line 285.

9. Line 157 the 14 species community has 14 unique community compositions?

9r. Yes, this is correct. There were 14 distinct, 14-species communities. The exact community compositions are listed in Table S3. We have added a reference to Table S3 here (line 295).

10. Line 250 how the absolute value of PSFs was calculated? I did not see relevant information in M&M part.

10r. Thanks, to clarify, we have added the following to line 332 in the Methods: ‘Because the mean of large positive and large negative PSF values can be zero, and therefore ‘mask’ PSF effects, we also calculated the absolute value of PSF values.’

11. Line 252 in 2018, most species experienced negative PSF and decreased plant growth by 10%, so 0.27 represents a negative value? I was wondering using “decreased” instead of “changed” could be a better choice and also clearer statement?

11r. We tried to be careful about distinguishing changes in the absolute value of PSF from increases or decreases in net PSF. To try to clarify, we have rewritten these sentences as follows (on line 151): ‘The absolute value of PSFs (0.27) indicated that two years of plant growth created soils that changed subsequent plant growth by 27%. However, because PSFs were both positive and negative, the net PSF effect was smaller (*i.e.*, a PSF value of -0.10 in 2018). Absolute PSF values reported across the literature tend to be larger (0.53)³⁶, but are mostly measured in greenhouse conditions that are known to exaggerate PSF values^{27,36,37}.’

12. Line 275-276 In fig. 3b, the overyielding was driven by selection effect, and why complementarity explained more overyielding than selection effect?

12r. In 3b (the 1997 B-P experiment), the complementarity effect was large across richness levels and the selection effect increased across richness levels. The selection effect does appear to explain the change in diversity effect across richnesses, but 1) this effect is smaller than the complementarity effect and the selection effects were not consistent between the 1997 and 2014 experiments. We are, therefore, more confident in the complementarity effect since it was consistent in both experiments.

We have tried to clarify and be more explicit about how PSFs change the selection and complementarity effects at the beginning of the Discussion as follows (line 128): ‘PSFs improved understanding of the magnitude and mechanism of the biodiversity-productivity relationship. In experimental communities, plants grew 118 g m⁻² more in diverse communities than in monocultures. This occurred because most plants grew better than expected from monocultures (*i.e.*, complementarity effects) and not because dominant species were over-represented in communities (*i.e.*, selection effects)¹¹. PSFs helped explain this pattern because most plants cultivated soils that decreased their own growth. Consequently, plants grew faster in communities, where they were surrounded by soils cultivated by other species than in monocultures, where they were surrounded by ‘self’ soils^{10,13,24}. This increases complementarity effects. Further, in Null simulation models, competition exaggerated monoculture growth differences among species. Consequently, plants that grew most in monoculture were predicted to be over-represented in communities (*i.e.*, selection effects). Negative PSF decreased these selection effects because dominant plants encounter higher proportion of ‘self’ soils than subdominant plants. The net effect of these changes was that PSF simulation models predicted 16.0 g m⁻² more biomass in diverse communities than monocultures, due to complementarity effects. This represented 14% of overyielding observed in experimental communities. PSF effects increased from 2017 to 2018 suggesting that PSF effects are likely to increase over time, though it is unlikely that PSFs would become a dominant determinant of overyielding. While

14% is a small portion of observed overyielding, results are important because they demonstrate diversity can increase productivity by suppressing plant disease. Results are also important because they help constrain the importance of other factors such as niche partitioning, which remain difficult to quantify²³.

13. Line 288 the difference between 60.7 and 70.7 is 10, but the later sentence said it was 10.3, please have a doublecheck.

13r. Thanks for catching this type-o. We have changed 10.3 to 10.0. (line 113)

Reviewers' comments:

Reviewer #2 (Remarks to the Author):

14. The authors describe results from three field experiments (one PSF experiment and two biodiversity experiments) as well as modeling that either included PSF effects or not. The authors found that the PSF effects explained 12-15% of the overyielding in biodiversity experiments, and that including PSF responses in models better predicted the underlying mechanisms (complementarity vs. selection effects).

I really like the fact that the authors did a PSF experiment in the field containing an unprecedented number of species and thousands of plots (albeit very small field plots), as this is where we need to do experiments for increased realism. It's also nice that surveys allowed for estimates of changes in PSF over time.

14r. Thanks, we agree that the magnitude of this field experiment will make it a cornerstone for PSF research. We have added comments about the change in PSF over time, as suggested by the reviewer below.

15. I am not a modeler and can therefore not comment on that aspect of the study although I can point out what is confusing to me with the hope that it may help the authors clarify aspects that may also confuse others. First, it is unclear to me what the authors mean when they say PSF models. Does the model predict the plant community total growth or the changes in growth across a plant diversity gradient that is supposedly due to PSF? If it's the former, it's doing a pretty poor job judging from the abstract and Figure 2 and severely underpredicts productivity. Also, it's not looking all that different from the null model without PSF in Fig. 2 and is this where the authors estimate the 12-15% contribution of PSF to the overyielding?

15r. Thanks for these comments. This is a recurring comment from reviewers and we have tried to address these important points in two ways. First, we have tried to be clearer in how we use the models. We now call the models 'plant community simulation models' that either include growth rates on different soils (PSF simulation models) or not (Null simulation models). We include the word 'simulation' to help distinguish these models from the mathematical partitioning of complementarity and selection effects. Second, we are now more explicit about the challenges inherent in predicting plant community composition and note that our predictive ability is similar to previous studies. Again, we have added the following to the Discussion to

address this point (line 191): In previous biodiversity-productivity experiments, species richness has explained 18% to 46% of variation biomass among communities^{8,40,41}. In our 2014 experiment, richness explained 12% of the variability in community biomass, suggesting large variability among communities, likely due to smaller plots and a shorter experiment duration. Despite this variability, our Null plant community model explained 12% of the variation in plant species biomass and our PSF model improved this correlation to 20%. Though correlations were not large (*i.e.*, >50%), results demonstrate that it is possible to predict species biomass in communities with similar accuracy reported for higher levels of organization (*i.e.*, community biomass vs. community richness) that are generally assumed to be easier to describe⁴²⁻⁴⁴.

We have also elaborated on our modelling approach in several locations to make clearer how models were used. At the end of the Introduction we note (line 51): ‘Our goal was to test whether field-measured PSFs could help predict plant growth in experimental plant communities with 1 to 16 plant species. To do this, we grew 16 species on soils cultivated by each species in the experiment (*i.e.*, a factorial PSF experiment). We used these soil-specific growth rates in a suite of plant community growth simulation models and compared model predictions to plant growth observed in experimental communities. We also ran the model with one growth rate for each plant species (*i.e.*, a Null simulation model with no PSF effects) to test how PSFs effected the model. To better understand the mechanisms determining how community biomass changes across species richness levels, we separated net biodiversity effects in each dataset (observed, Null, PSF) into complementarity and selection effect components^{11,33}.’

In the Methods we now note (line 347): ‘PSF experiments describe plant growth on soils cultivated by different species, but do not describe how plants grow in communities. To assess how these PSFs are likely to affect plant growth in communities, we use plant community simulation models with and without PSF effects to predict plant biomass and we compare model predictions to plant biomass observed in experimental plant communities. Broadly, these models allow each plant in a community to grow from seed at rates determined from the PSF experiment. Plant growth is eventually limited by a carrying capacity.’

16. Other than that, I found it relatively easy to follow, but I wonder if the Discussion would be clearer by organizing it into subheadings making some main points?

16r. Thanks, this is a great suggestion. We have added the following two subheadings to the Discussion: ‘Insights from a large factorial PSF field experiment’ (line 161), and ‘Quantifying PSF effects on plant growth in communities’ (line 190)

Below are some more specific comment that I hope are useful when the authors revise the current manuscript.

17. • Abstract. The third bullet point is confusing without the proper background. For example, while the PSF model is better as it predicts more overyielding due to complementarity, what do we compare that to so that we understand that the model without PSF incorrectly assign the overyielding to selection effects? Both seem pretty bad if we are comparing that to experimental findings where complementarity explained 185% of the overyielding. Also, under bullet point four, I wouldn’t say that the PSF model improved the magnitude of predictions as it only went from 17 to 27% whereas the real (?) value was 185%. Am I missing something here or do both

models severely underpredict overyielding? Or, does the PSF model predict the proportion explained by PSF, but if that is the case, I do not understand why the PSF models predict complementarity (lines 315-317).

•17r. Thanks. These are important points to clarify. We have had to cut the abstract length in half, so we address this comment (that PSF explained a small portion of the very large 185% effect) directly at several points in the paper and we believe doing so has improved the paper by placing the results in the context of other factors that also determine the biodiversity-productivity relationship.

In the Abstract (line 10) we note: ‘Though this effect alone was modest, it helps constrain the role of factors, such as niche partitioning, that have been difficult to quantify.’

We address this again in the beginning of the Discussion (line 142) as follows: . PSF effects increased from 2017 to 2018 suggesting that PSF effects are likely to increase over time, though it is unlikely that PSFs would become a dominant determinant of overyielding. While 14% is a small portion of observed overyielding, results are important because they demonstrate diversity can increase productivity by suppressing plant disease. Results are also important because they help constrain the importance of other factors such as niche partitioning, which remain difficult to quantify²³.’

We address this point again with the concluding sentences (line 214) as follows:’ Even though this effect is likely to increase over time, it is likely to remain modest relative to 100% to 200% increases in productivity across species richness treatments. Yet, demonstrating a 14% PSF effect is important because it quantifies how diversity can increase productivity in communities by suppressing plant disease. It is also important because it helps constrain the role of other factors (i.e., niche partitioning) in biodiversity-productivity relationships^{25,52,53}. Future research that quantifies and integrates niche partitioning with PSF and other effects can be expected to improve predictions of the effects of species loss on plant community productivity and resilience with implications for biofuel production and conservation^{6,54}.’

18. Line 37. Correct spelling of overyielding.

18r. Thanks, we have made this change.

19. • Lines 40-41. I think you mean mutualists, not symbionts as “symbiont” just mean living in close proximity but says nothing about the nature of the interaction. Also, I can see species-specific mutualists with legumes, but given that this is a grassland community, most plants will associate with AMF that are not particularly specific so I see no obvious reason why they would be more abundant (or beneficial) in monocultures as it all depends on host quality.

19r. Thanks, this is a good point. We have changed ‘symbionts’ to ‘mutualists’ as suggested. (line 29)

20. • Lines 73-76. Shouldn't one factor also be PSF in your model?

20r. We are referring to simulation models, not statistical models. We have changed the text throughout the paper to address this point.

21. • Line 102. What was the distance between plots?

• Lines 115-119. This fallow period is a lot longer than many greenhouse experiments use, which

may be one reason for why significant PSF effects were only observed in 15% or so of the treatments? That may be worth commenting on in the Discussion?

21r. We now note that plots were immediately adjacent to one another (line 261). The fallow time was longer, but this was during the winter when there was no plant growth.

22. • Line 121. Seeded? If so, by using the same seed density as in Phase 1?

22r. We have added that Phase II seeding rates matched Phase I seeding rates. (line 277)

23. • Line 135. Probably a good idea to also outline what the “maximum” S and O refer to.

23r. To address this, we have added (line 324) ‘and max(S,O) selects the larger of S and O’

24. • Lines 140-146. I find “soil type” to be distracting here as all experiments were conducted on one soil type. I would prefer soil training. Also, does the 5-9 other replicates mean for each other species the focal plant was grown on, making the total other reps (5-9) x 15?

24r. Thanks, this is a good suggestion. We have changed ‘soil type’ to ‘soil training type’ throughout the paper. Yes, each plant was grown on 5-9 replicates for each soil cultivation type. As a result, soil*species PSF values are derived from 27-35 ‘self’ replicates and 5-9 ‘other’ replicates and species PSF values are derived from 27-35 ‘self’ replicates and ~105 ‘other’ replicates.

25. • Lines 149-150. Given that you have 16 species and you ran t-tests on all of them, how do you control for false positive if the individual error rate is 0.05? Seems like at least one could be significant just by chance if not controlled for. Then again, if you do control for multiple tests it becomes so conservative as to become meaningless so not sure what the best approach is here.

25r. We do not adjust for the 16 tests. Rather than focus on the statistical significance of each PSF value, we focus on the biological significance of PSFs by comparing model predictions to observed plant growth. We are now clearer that the focus of the paper is on using plant growth rates from the PSF experiment in simulation models and that we present PSF values simply as a general summary that is easily compared to values in other studies. We agree with the reviewer that a bonferonni type adjustment would be too conservative. More importantly, it wouldn’t affect our results or conclusions, since we use growth rates and not PSF values in our models.

26. • Lines 247-250. Does this mean that only 15% (36/240) actually showed a significant PSF and that the great majority were neutral? Also, 13+23 is 36, not 39 unless I am missing something here.

26r. Thanks for catching this, 39 was a type-o and should be 36 (line 78). Yes, 85% of the values had confidence intervals that overlapped zero. Again, this wasn’t particularly important to our study because we used mean growth rates in our models to test for biologically significant effects on plant growth regardless of the statistical significance of the PSF value.

27. • Line 269. How can you have selection effects in a community that contain all the 16 species given that there is zero change that a particularly productive species wouldn’t be selected? I must be missing something here.

27r. Selection effects in communities, even when all species are present, are common. Selection effects essentially summarize how competition exaggerates competitive inequalities. Since competitive effects can be exaggerated in a community with all component, species, this effect can occur when all species are present. For example, a plant with greater than average

monoculture biomass would be expected to have a competitive advantage. If that species represents more biomass in a community than one would predict from its relative monoculture yield, then the difference between expected and observed yield is counted as a selection effect. This process is not sensitive to whether or not all component species are present in a community.

28. • Lines 303-304. The fact that PSF explain 12-15% of overyielding in biodiversity experiments seems “abstract-worthy”. Also, where does that estimate come from? It’s not anywhere in the Results so is it the difference between the null model and PSF model?
28r. This is a good point. The 12-15% came from the 1997 and 2014 experiments, but we now summarize those two experiments and just say 14% now. We now make clear in the results that the mean effect in the two experiments was 118 g m⁻² (line 101). In the Discussion we make clear that the two experiments resulted in 118 g m⁻² due to overyielding and that the PSF model predicted 16 g m⁻² overyielding (i.e., 14%). This is summarized in the first paragraph of the Discussion (line 128).

29. • Lines 328-330. I do not understand why the PSF model would decrease the biomass of positive PSF. If plants experience positive PSF, shouldn't the model promote their biomass relative to a null community without PSF? What am I missing here?

29r. This is an important point and one we have detailed in previous manuscripts. It is a common misconception that negative PSFs are ‘bad’ for plant growth and positive PSFs are good for plant growth. By definition, a plant with a negative PSF grows more on ‘other’ than on ‘self’ soils. As a species with a negative PSF grows in a community, it encounters more and more ‘other’ soil and is, therefore, expected to overyield. In contrast, a plant with a positive PSF grows more on ‘self’ than ‘other’ soil. As this plant grows in a community, it encounters less and less ‘self’ soil and therefore underyields. We describe this process in the Introduction (lines 26-35) and we now describe this process more clearly in the first paragraph of the Discussion (line 128).

30. • Lines 334-336. Do the other studies that are reference also report absolute PSF? I am more familiar with PSF means.

30r. We have used this metric, though it is not widely used. It is informative, because as we now note in the paper, mean values can easily mask PSF effects. For example, two species with a large positive and a large negative PSF, would have a net neutral feedback, yet it would be incorrect to describe PSF as unimportant for these two species. The absolute value of PSF describes the extent of PSF regardless of the sign of the value. We address this point on lines 70, 150 and line 331.

31. • Fig. 1a. It would be good to use vertical lines to separate the functional groups. Also, could the authors do some sort of analysis to see if there is any relationship between phylogenetic relatedness and PSF? For example, was PSF more negative when grown after a more closely related species as would be predicted if pathogens are species specific and respond to similarities in traits (if traits are phylogenetically conserved that is)? It could help us get to underlying mechanisms to better understand and predict PSF. Could the authors correlate pairwise phylogenetic distance among plants with PSF?

31r. Thanks for this suggestion. We have added vertical lines to separate the functional groups in Fig. 1b.

modeler like myself....

32r. Thanks. We have rewritten Figs 2 and 3 legends to clarify.

Fig. 2: Observed and predicted plant biomass in experimental plant communities with one to 16 plant species. Plant community growth simulation models either with (PSF) or without (Null) plant-soil feedback effects predicted that biomass would increase with species richness (i.e., overyield). However, PSF simulation models correctly predicted this effect was caused by complementarity effects and Null models incorrectly predicted this effect was caused by selection effects (Fig. 3). The overyielding predicted by PSF simulation models represented 14% of the overyielding observed in the two biodiversity-productivity experiments. Each point represents total aboveground biomass in one community type ($n = 55$ or 63 for the 1997 and 2014 experiments, respectively). Large values from six outlier plots are not shown but were included in analyses. In each dataset, biomass increased with species richness ($P < 0.05$; see Results for details).

Fig. 3: Observed (a and b) and predicted (c and d) biomass responses to species richness. Net effects (black symbols) were separated mathematically into complementarity effects (red) and selection effects (blue). Plant community simulation models that included plant-soil feedbacks correctly predicted that plant growth would increase with richness due to complementarity effects (c; red line) while the same models without plant-soil feedback effects incorrectly predicted positive selection effects (d; blue line) and negative complementarity effects (d; red line). Data from 63 (a) or 55 (b) different replicated plant communities. In each

dataset, the net biodiversity effect increased with species richness ($P < 0.05$; see Results for details).

33. • Fig. S1. I don't see any blue here.

33r. Thanks we have changed this to green.

Reviewer #3 (Remarks to the Author):

34. This manuscript describes results from a group of field experiment that estimated how much plant-soil feedbacks contributed to a positive diversity-productivity relationship in a grassland community. Overall, plant soil feedbacks (performances of plant species in own versus other soil) were negative and varied considerably for each plant/soil combination. The diversity productivity relationship was strongly positive and driven by complementarity effects. Plant community growth models that either included or did not include measured plant-soil feedbacks were used to predict total biomass in the experimental diversity communities. Plant-soil feedbacks predicted about 15% of the biodiversity-productivity relationship and predicted complementarity as the predominant mechanism. However, the models did not predict biomass in a longer-term plant-soil feedback experiment.

First, I'd like to commend the authors on pulling off an ambitious, well-designed experiment that contained almost 3,000 (amazing!!) plots that were maintained for four years. While several hundred diversity-productivity experiment have been conducted, the mechanisms underlying diversity-productivity relationships have remained elusive. Plant-soil feedbacks certainly offer one potential mechanism that can be more readily tested than other mechanisms, such as niche partitioning. While a handful of other experiments have linked plant-soil feedbacks to diversity-productivity relationships before, to my knowledge this is the first two-phase field experiment – which likely increases realism relative to greenhouse experiments. It is also very well-replicated (a common failing in plant-soil feedback experiments), increasing confidence in experimental results.

34r. Thanks, we feel the three reviewers have all recognized the scale and vigor / value of the research.

I have a few comments that may improve the manuscript:

35. 1. It is likely that plant-soil feedbacks vary with diversity (e.g., Hawkes et al. 2013) and the frequency/density of conspecific individuals (e.g., Chung and Rudgers 2016). Measuring these differences and incorporating them into the plant community growth models are certainly outside of the scope of this experiment, but I do think the issue should be addressed in the discussion. With more fine-scale information, plant-soil feedback may become a better (or worse) predictor of plant biomass in diverse communities.

Chung, Y. A., and J. A. Rudgers. 2016. Plant–soil feedbacks promote negative frequency dependence in the coexistence of two aridland grasses. *Proc. R. Soc. B* 283:20160608.

Hawkes, C. V., S. N. Kivlin, J. Du, and V. T. Eviner. 2013. The temporal development and additivity of plant-soil feedback in perennial grasses. *Plant and Soil* 369:141–150.

35r. Thanks for this comment. This is an excellent point. We have added the following to line 205 in the discussion to address this point An implication of the poor correlation between the new and old data is that inference about the effects of PSF on plant community development are likely to be time- or site-dependent^{46–49}

36. 2. Glyphosate was used to remove unwanted plants when the experiment was established and after the conditioning phase of the plant-soil feedback experiment. Does glyphosate have any known effects on soil microbial communities? This may be a concern for measuring feedbacks if it disproportionately affects mutualists or pathogens – although effects are equal across the plant-soil feedback experiment. However, the glyphosate was not reapplied to the biodiversity experiment. If glyphosate influenced soil microbes, it may influence correlations between biomass predicted in plant-soil feedback models and the observed biomass in the diversity experiment.

36r. Thanks, this is a good point. To address this, we have added the following to line 277: Glyphosate application may affect mycorrhization and therefore decrease positive PSF⁵⁶, but it was critical to ensure that all Phase I plants were killed because resprouting plants have the potential to create large, false positive PSFs.

37. 3. In the results from the plant-soil feedback experiment, there is evidence that plant-soil feedbacks become more negative (on average) through time. If they continue to change, what effect does this have on model predictions of community biomass?

37r. Thanks, this is a good point. We now address this in the abstract (plants created soils that changed subsequent plant growth by 27% and that this effect increased over time; line 7) and the Discussion (lines 142): ‘. PSF effects increased from 2017 to 2018 suggesting that PSF effects are likely to increase over time, though it is unlikely that PSFs would become a dominant determinant of overyielding.

38. 4. Null models did not predict the observed complementarity effect, but plant-soil feedback models did, suggesting that plant-soil feedback models do a better job explaining community biomass than null models, despite both predicting the positive diversity-productivity relationship. Can null models produce complementarity effects? It seems like the answer is no since there is no chance for another mechanism (niche partitioning) to produce complementarity effects in the plant-soil feedback experiment, but I may be missing something? Also, how do the models hold up to predicting plant community composition in diverse communities? Is one better than the other?

38r. Yes, this is correct. There is no complementarity mechanism in the Null models. We are using the Null models as a way to measure the complementarity effect produced by PSFs. We address this in the methods as follows (lines 347): ‘PSF experiments describe plant growth on soils cultivated by different species, but do not describe how plants grow in communities. To assess how these PSFs are likely to affect plant growth in communities, we use plant community simulation models with and without PSF effects to predict plant biomass and we compare model predictions to plant biomass observed in experimental plant communities. Broadly, these models

allow each plant in a community to grow from seed at rates determined from the PSF experiment. Plant growth is eventually limited by a carrying capacity. The best-performing discrete plant community simulation models in a similar previous study were used (i.e., the ‘logistic species-level-K model’ and the ‘logistic constant-K model’)^{59,60}. In this logistic growth simulation model, species-conditioned soils ‘grow’ as a function of plant biomass, plant species growth rates, and a plant-to-microbe conversion factor. Plant growth rates are a function of the proportion of different soil training types present. To prevent run-away growth, biomass is limited by a carrying capacity, which can be either unique to a species or to the community. Null model simulations are the same except that they include only one soil training type and one plant growth rate (SI Appendix). The Null version of these models does not include a complementarity mechanism, but they can produce selection effects. ‘

Minor edits:

39. Figure 2 caption: no need for “a” in “in a new”

39r. Thanks, we have made this change.

REVIEWERS' COMMENTS:

Reviewer #1 (Remarks to the Author):

All of my comments have been adequately addressed and I recommend this manuscript for publication.

Reviewer #2 (Remarks to the Author):

I am satisfied with the changes the authors have made and the manuscript is now a lot clearer. I only have some minor comments.

- Lines 9-10. Isn't it a bit strong to say that diversity maintains productivity by suppressing disease given the low explanatory power of PSF (14%)? The main effect is still complementarity, isn't it?
- Line 58. Remove one "with".
- Line 90. Still soil types here so may want to change to soil training history or soil training type? Again in line 170.
- Line 115. Perhaps to be super clear add (without PSF estimates) or something like that after Null simulation models?
- Lines 210-212. That's setting the bar pretty low, isn't it?
- Lines 278. I assume the method used was based on rooting depth or density or something like that? I also assume that the different methods did not have any effect on the results?
- Line 329. The "but" threw me off. If it is an added benefit, shouldn't it be "and"?
- Figure 1. You may want to add the level of significance the asterisks represents.

Reviewer #3 (Remarks to the Author):

Thank you to the authors for carefully addressing my comments. I have no further comments on the manuscript.